# Performance of Adaptive Bit-Interleaved Polar Coded Modulation in FSOC System

**Qingfang Jiang** [1], **Zhi Liu** [2,3,*], **Haifeng Yao** [4], **Zhonglin Luo** [5], **Xin Zhang** [5], **Shutong Liu** [1], **Chenming Cao** [5], **Gang Jing** [5], **Hao Li** [5] and **Peng Lin** [5]

1   School of Electronics and Information Engineering, Changchun University of Science and Technology, Changchun 130022, China; jqf@mails.cust.edu.cn (Q.J.); liushutong0920@sina.com (S.L.)
2   National and Local Joint Engineering Research Center of Space Optoelectronics Technology, Changchun University of Science and Technology, Changchun 130022, China
3   Institute of Space Optoelectronics Technology, Changchun University of Science and Technology, Changchun 130022, China
4   School of Optoelectronics, Beijing Institute of Technology, Beijing 100081, China; scifeng@bit.edu.cn
5   School of Optoelectronics Engineering, Changchun University of Science and Technology, Changchun 130022, China; luozl@mails.cust.edu.cn (Z.L.); 2021100384@mails.cust.edu.cn (X.Z.); chenming@mails.cust.edu.cn (C.C.); jg@mails.cust.edu.cn (G.J.); lihao@mails.cust.edu.cn (H.L.); lp@cust.edu.cn (P.L.)
*   Correspondence: liuzhi@cust.edu.cn; Tel.: +86-158-4408-8833

**Abstract:** This paper proposes an adaptive bit-interleaved polar coded modulation (A-BIPCM) method based on minimum logarithmic upper bound weight (MLUW). It is designed to reduce the fading effects and long string of bit error interference caused by atmospheric turbulence in free-space optical communications (FSOC). To assess the effectiveness of this method across turbulent channels of varying intensities, we conducted an evaluation of the bit error rate (BER) performance of polar codes in turbulent channels. The results demonstrate significant performance improvements provided by the A-BIPCM method compared to conventional polar code encoding and decoding. Specifically, under weak, moderate, and strong turbulence conditions, the A-BIPCM method achieves performance gains of 0.96 dB, 1.66 dB, and 1.35 dB, respectively. Additionally, an experimental verification platform for FSOC employing intensity modulation direct detection (IM/DD) with an atmospheric turbulence simulation channel, is established in this work. When the optical power of the detector is $-16$ dBm, the traditional polar code encoding and decoding performance at BER $= 2.36 \times 10^{-5}$, whereas the A-BIPCM scheme exhibits a significantly higher performance at BER $= 2.11 \times 10^{-6}$. The BER has been improved by representing an order of magnitude.

**Keywords:** free-space optical communication; bit-interleaved polar code modulation; minimum logarithmic upper bound weight

## 1. Introduction

Free-space optical communication (FSOC) has the advantages of high bandwidth, low cost, and large transmission capacity. It is considered to be one of the solutions to the next generation of wireless communication problems and can meet growing communication needs [1]. However, in fading channels, the primary cause of beam fading is the fluctuation of the optical refractive index induced by atmospheric turbulence. This turbulence-induced fluctuation of the refractive index can give rise to various phenomena, including light intensity flickering, beam expansion, and beam drift, effects that can lead to the occurrence of code errors over extended transmission periods. To mitigate the impact of atmospheric turbulence, forward error correction (FEC) techniques are extensively employed in FSOC systems. Prominent examples of FEC codes utilized in FSOC include turbo codes, low-density parity-check (LDPC) codes, and polar codes [2]. However, FEC techniques may not be entirely effective in dealing with long strings of bit errors caused by channel fading. In

order to further improve the performance of the system, bit-interleaved coded modulation (BICM) is introduced into FSOC as a communication mode. Initially proposed by Zehavi, BICM is a scheme that combines error correction coding, interleaving, and modulation in a serial concatenation manner [3]. BICM offers lower computational complexity and improved system reliability in the channel [4].

Theoretical analysis has proven that polar codes can achieve the Shannon limit with linear encoding and decoding complexity, presenting new possibilities in the field of channel coding [5]. In 2016, Hessam Mahdavifar proposed a polar coding scheme integrated with BICM for reliable communication over multiple channels. Although this scheme demonstrated the same error exponent as Arikan's polar codes and achieved multi-channel capacity, it lacks experimental validation [6]. In 2018, Saha introduced error correction techniques to integrated polar codes, presenting a high-performance scheme known as bit-interleaved polar coded modulation with iterative decoding (BIPCM-ID). BIPCM-ID further improved performance by increasing the number of iterations and achieving higher diversity between transmitted bits. However, its application in fading channels remains unexplored [7]. In 2020, Niu Kai et al. established a comprehensive framework analyzing the theoretical performance of polar codes in block fading channels. Through the analysis of the polar spectrum, they revealed the explicit relationship between diversity order and codeword weight, yet this theoretical framework has not been applied to BIPCM systems [8]. In 2022, Hyosang Ju et al. proposed an interleaving method that achieves full diversity at a rate of 0.5 in the presence of two fading blocks. This indicates the potential of BICM techniques to optimize communication links in fading channels [9]. Also in 2022, Jiang Lun's team introduced the multi-aperture transmission and BICM techniques into FSOC, aiming to suppress atmospheric turbulence. By leveraging a random interleaver, they improved the minimum Euclidean distance between codewords, enhancing the link's resistance to fading. This research suggests that further exploration of advanced BICM techniques could strengthen the suppression capability against atmospheric turbulence and improve the environmental conditions of space links [10]. Traditional polar coding methods have limitations in effectively solving the problem of a long string of bit errors.

We introduce a scheme of adaptive bit-interleaved polar coded modulation (A-BIPCM) to address the issue of a long string of bit errors caused by turbulence in FSOC systems. The A-BIPCM scheme employs the analysis of pairwise error probability (PEP) and polar spectrum (PS) to evaluate the minimum logarithmic upper bound weight (MLUW) approach, we use MLUW to construct the adaptive interleaving criterion. Simulations were conducted to evaluate the bit error rate (BER) performance of both conventional polar codes and the A-BIPCM method across various levels of turbulent channel intensities. The A-BIPCM scheme achieves a 0.96 dB improvement in weak turbulence, a 1.66 dB improvement in medium turbulence, and a 1.35 dB improvement in strong turbulence. Furthermore, experimental validation confirms that the adoption of the A-BIPCM technique effectively enhances the transmission quality of FSOC systems operating in weak turbulence channels. Which proves to be effective in combating a long string of bit errors induced by turbulence.

## 2. Adaptive Bit-Interleaved Polar Coded Modulation (A-BIPCM)

### 2.1. Construction of FSOC System and Channel Construction Based on Adaptive Bit-Interleaved Polar Code Encoding

The proposed Principle block diagram of an FSOC system based on bit-interleaved polar code coding is illustrated in Figure 1. The system operates with an input source sequence of length $k$, consisting of an information sequence and $m$ cyclic redundancy check (CRC) transmission bits. The parallel sequences to be encoded are converted from serial to parallel format by the encoder. These sequences $x_1^N = x_1, x_2, \cdots x_N$ undergo parallel polar coding, which is divided into $r$ subchannels. The encoded sequences are then passed through an interleaver $s(t)$, which reorders the bits to enhance error correction capabilities. The information sequence is divided into blocks with a length equal to the length of the interleaver according to the turbulence intensity. Adjacent bits of the coded bit sequence

can be transmitted at intervals. After interleaving, the sequences are mapped and amplified using an erbium-doped optical fiber amplifier (EDFA). Subsequently, the amplified signals $S(t)$ are transmitted through a fiber collimator (FC) into a fading channel. At the receiver end, the signal model is as follows:

$$y(t) = \eta \cdot I(t) \cdot S(t) + n(t) \tag{1}$$

In this formula, the received signal is denoted by y(t), which is the signal detected by the receiver. $\eta$ represents the detector responsivity of the photodetector, and we assume that $\eta$ is equal to one for simplicity. The transmitted signal is represented by $S(t) \in \{0, 1\}$. $I(t)$ accounts for the intensity attenuation caused by atmospheric turbulence, which affects the received signal strength. $n(t)$ represents the zero-mean Gaussian white noise experienced by the receiver, which contributes to the overall noise in the received signal. After receiving the signal, the received sequence undergoes demodulation and mapping processes. Subsequently, the deinterleaving operation is performed to reorder the sequence to its original order. The deinterleaved sequence is then delivered to the decoder [11].

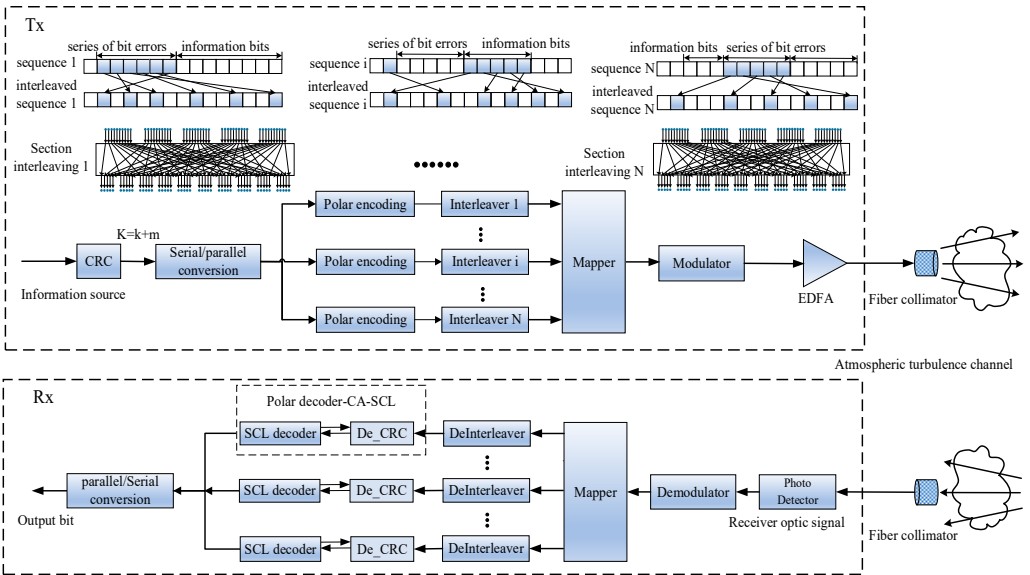

**Figure 1.** Schematic diagram of FSOC system with bit-interleaved polar coding modulation.

In this work, we employ the Monte Carlo method to construct a simulation channel for atmospheric turbulence characterized by Gamma-Gamma random numbers. This model aims to provide a more precise depiction of the light propagation phenomena, encompassing both large-scale and small-scale irradiance fluctuations [12]. The Gamma-Gamma model employed in this research is defined as follows:

$$f(\langle I \rangle) = \int_0^\infty \frac{2(\alpha\beta)^{\alpha+\beta/2}}{\Gamma(\alpha)\Gamma(\beta)} \langle I \rangle^{\alpha+\beta/2-1} K_{\alpha-\beta}(2\sqrt{\alpha\beta\langle I \rangle}) dI \tag{2}$$

In the formula, normalization can be expressed as $\langle \cdot \rangle$, and the normalized light intensity can be defined as $\langle I \rangle = I/E(I)$ [13]. $C_n^2$ represents the structural constants of the atmosphere. $L$ represents the free space link distance, $\kappa = 2\pi/\lambda$ represents the space wave number, $\lambda$ represents the wavelength of light, $I$ represents the variable, and $\Gamma(\cdot)$ is the gamma function. $K_{\alpha-\beta}(\cdot)$ is the Bessel function of the second kind of $\alpha - \beta$ order. $\alpha$ and $\beta$ are related to atmospheric conditions, which are given by:

$$\alpha = \left[\exp\left(\frac{0.49\sigma_{Rytov}^2}{\left(1 + 1.11\sigma_{Rytov}^{12/5}\right)^{7/6}}\right) - 1\right]^{-1}, \beta = \left[\exp\left(\frac{0.51\sigma_{Rytov}^2}{\left(1 + 0.69\sigma_{Rytov}^{12/5}\right)^{7/6}}\right) - 1\right]^{-1} \qquad (3)$$

The progressive state of the atmosphere can be characterized by the Rytov variance [14]:

$$\sigma_{Rytov}^2 = 1.23C_n^2 k^{7/6} L^{11/6} \qquad (4)$$

The Rytov variance can measure the intensity of atmospheric turbulence, and the atmospheric coherence constant can be obtained based on the Rytov variance:

$$r_0 = 1.89\left(k^{-5/6} L^{5/6} \sigma_{Rytov}^2\right)^{-3/5} = \left(0.423k^2 C_n^2 L\right)^{-3/5} \qquad (5)$$

*2.2. Adaptive Bit-Interleaving Polar Coding Modulation Method Based on Approximate Error Probability Upper Bound*

In fading channels, atmospheric turbulence can cause a long string of bit errors, leading to an increased system transmission BER. To address this issue, this paper proposes a new method of adaptive bit-interleaved polar coding modulation (A-BIPCM) The encoded sequence is sent to the interleaver, and the polarization spectrum (PS) and the pairwise error probability (PEP) upper bound are used to study the error correction code scheme. An upper bound on the average bit error probability of the coding system is obtained, and the interleaver depth is varied based on a numerical evaluation of the performance bound. The core principle of A-BIPCM is to adaptively adjust the parameters of bit-interleaving and polar coding based on these error probability upper bounds. The introduction of A-BIPCM effectively mitigates the interference caused by a long string of bit errors induced by atmospheric turbulence, thereby enhancing the reliability and performance of the system; this work designs an interleaver tailored to different turbulence intensities. It upsets and changes the order of the original binary coding sequence, interleaving the erroneous bits between the information bits. Additionally, the original binary coding sequence is split into corresponding positions of the error correction code group, resulting in an interleaved sequence group.

The average PEP is a measure of the distance between all-zero codewords and non-zero codewords in a polar code. It quantifies the likelihood of pairwise error events occurring between different codewords. In this work, the average PEP is denoted by the symbol $P_N^{(i)}(d)$. The polar spectrum $A_N^{(i)}(d)$ is a collection of weighted enumerators, each associated with a specific Hamming weight [15]. It is used to count the number of pairwise error events and evaluate the performance of polar subcodes. The polar spectrum is defined as a collection of weight distributions $\left\{A_N^{(i)}(d)\right\}, d \in [1, N]$, where $d$ represents the Hamming weight of non-zero codewords. Considering a code length of N and assuming transmission over a fading channel, the minimum weight enumerator is used to approximate the upper bound mentioned above. The MLUW metric is expressed by the following formula [16]:

$$MLUW_N^{(i)} = L_N^{(i)}\left(d_{\min}^{(i)}\right) - d_{\min}^{(i)} \ln \frac{E_S}{N_0} \qquad (6)$$

In this work, the upper bound of the minimum Hamming distance of the polar subcode $D_N^{(i)}$ is denoted by the symbol $d_{\min}^{(i)}$. For a given polar subcode $D_N^{(i)}$, under a specific SNR $\gamma = E_S/N_0 \gg 1$, given an index set $\mathcal{A} \subseteq [1, N]$. Given the fixed configuration $(N, K, \mathcal{A})$, the approximate BER upper bound of the polar code in the fading channel is expressed as $P_e(N, K, \mathcal{A})$ [17]:

$$P_e(N, K, \mathcal{A}) \leq \frac{1}{2}\sum_{i \in \mathcal{A}} A_N^{(i)}\left(d_{\min}^{(i)}\right)\left(\frac{E_S}{N_0}\right)^{-d_{\min}^{(i)}} \qquad (7)$$

By setting the upper bound of the BER to $3.8 \times 10^{-3}$ in the above formula, the corresponding SNR threshold $\gamma_1$ can be determined. This threshold aids in determining whether interleaving should be employed in the transmission process. This paper divides the transmission process into blocks, with each block spanning a duration of 2 s. Different interleaving methods are selected for each channel block based on the intensity of atmospheric turbulence. However, it should be considered that interleaving introduces additional delay and complexity to the system. Hence, a threshold judgment is necessary to determine whether interleaving should be applied. The average SNR is recorded using the coherence time $\tau_0$ interval and is calculated from the SNR $\overline{\gamma}$. The following formula represents the SNR of a signal with interference noise. It is calculated by comparing the received signal with the signal generated by the transmitter.

$$SNR_{\text{sig}} = \frac{|\mathbf{r}_{AB}(t)|^2}{|\mathbf{r}_{AB}(t) - (\rho_A S(t) + n_A(t))|^2} \tag{8}$$

$|\mathbf{r}_{AB}(t)|^2$ is the total power of the received signal and represents the squared amplitude of the signal. $|\mathbf{r}_{AB}(t) - (\rho_A S_A(t) + n_A(t))|^2$ represents the power of noise and interference signals, which represents the resistance of the signal source to noise and interference signals. $r_{AB}(t)$ represents the signal that reaches the receiving end B after being transmitted through the channel. $\rho_A$ is the amplification coefficient of the signal. $\rho_A S_A(t)$ is the signal of the signal source at the transmitting end, $n_A(t)$ is the received noise signal.

By determining specific threshold conditions, it is possible to assess whether interleaving operations are necessary to optimize system performance. For the received signal, the SNR level of the current channel block is estimated to evaluate its quality. Within the segment, the average signal-to-noise ratio $\overline{\gamma}$ is recorded every coherence time $\tau_0$, and the signal-to-noise ratio threshold is set to $\overline{\gamma}_1 = 8$ dB. The signal-to-noise ratio judgment principle is as follows:

$$\begin{cases} \overline{\gamma} < \overline{\gamma}_1 = 8 \text{ dB, Marked as 1} \\ \overline{\gamma} \geq \overline{\gamma}_1 = 8 \text{ dB, Marked as 0} \end{cases} \tag{9}$$

If the average SNR over a certain coherence time $\tau_0$ is lower than the SNR threshold $\gamma_1$, the corresponding sub-segment is marked as having a bit error. Conversely, if the average SNR exceeds the threshold, for the number of $2/\tau_0$ times that needs to be detected, the sub-segment is considered error-free. In this work, the number of times that the threshold condition is met and the sub-segment is marked as having a bit error is denoted as $Q$. Consequently, a relationship can be established between the total number of detections and the number of bit errors in the segment.

$$\begin{cases} P_1 = \frac{Q}{\frac{2}{\tau_0}} = \frac{1}{2}Q\tau_0 \geq P_{th}, \text{ Use interweaving} \\ P_1 = \frac{Q}{\frac{2}{\tau_0}} = \frac{1}{2}Q\tau_0 < P_{th}, \text{ No interleaving} \\ \qquad P_1 + P_0 = 1 \end{cases} \tag{10}$$

The decision to use interleaved polar codes to enhance error performance is based on the cumulative error detection probability. If the cumulative error detection probability surpasses a certain threshold $P_{th}$, interleaving operations are employed. On the other hand, if the cumulative error detection probability does not exceed the threshold $P_{th}$, there is no need for interleaving, and the received demodulation sequence can be directly passed to the decoding module for error correction. The selection of the threshold can be adjusted according to specific requirements, aiming to strike a balance between achieving optimal performance and managing system complexity. By fine-tuning the threshold, the trade-off between error correction capabilities and computational overhead can be optimized.

Interleaved polar codes are constructed recursively through the structure $x + y|y$. Let $C_1$ and $C_2$ be two linear codes of the same length, which $C_1 + C_2|C_2$ will be defined as:

$$C_1 + C_2|C_2 = \{[x + y]|x \in C_1, y \in C_2\} \tag{11}$$

Polar codes can be described by the following recursive equation:

$$C_{m,j} = C_{m-1,2j} + C_{m-1,2j+1}|C_{m-1,2j+1} \tag{12}$$

where $m = 1, \dots, M$ and $j = 0, \dots, 2^{M-m} - 1$, the initial condition is $C_{0,j} = \{0, 1\}, j \in A$ and $C_{0,j} = \{0\}, j \in A^c$. The length of the polar code is $N = 2^M$ represented by $C_{M,0}$. Interleaved polar codes are constructed by $C_{m-1,2j}$, inserting an interleaver at the output of each encoder. The interleaver can be expressed as a permutation matrix $\Pi$, which $C\Pi$ will be defined in this article as:

$$C\Pi = \{x\Pi \,|\, x \in C\} \tag{13}$$

where $C\Pi$ represents the new sequence obtained by permuting all codeword sequences C using an interleaver $\Pi$. Therefore, the interleaved sequence can be described by the following recursive equation

$$C_{m,j} = C_{m-1,2j}\Pi_{m-1} + C_{m-1,2j+1}|C_{m-1,2j+1} \tag{14}$$

## 3. Performance Simulation Results and Discussion of A-BIPCM under Different Turbulence Conditions

Based on the method mentioned above, this paper conducts simulation analysis on key parameters such as turbulence intensity, decoding width, and before and after adaptive interleaving to evaluate the bit error performance of A-BIPCM under atmospheric turbulence. To achieve this goal, this article introduces CRC into the traditional SCL to improve decoding performance and uses CRC check to verify the accuracy of the decoding results, which has proven potential in short block length delay-constrained systems. In order to provide a comprehensive performance evaluation and take into account the application requirements in the actual system, during the simulation process, the following parameters were set: code length N = 2048, code rate R = 0.5, wavelength $\lambda = 1550.116$ nm, CRC check code length is 8, and the number of iterations is set to $10^6$. In addition, simulation analysis was also conducted on different list sizes, which were, respectively, set to L = 1, L = 4, and L = 8.

In this work, a comparison is made between the proposed A-BIPCM scheme and the conventional polar code encoding and decoding. The comparison is based on simulations that analyze the change in BER with SNR under weak turbulence, medium turbulence, and strong turbulence conditions (as shown in Figure 2a–c). The Rytov variances under the three turbulence conditions are 0.2, 1.6, and 3.5, respectively. The A-BIPCM scheme is discussed, particularly focusing on the differences observed under different code rates and the list size. To investigate the performance gap between polar codes and the Shannon limit in turbulent channels from an information theory perspective, the Shannon limit curve is also plotted for each turbulent condition, considering a code rate of 0.5. The simulation results indicate that at the BER = $1 \times 10^{-5}$, code rate $R_2 = 0.75$, the proposed A-BIPCM scheme achieves an improvement compared to conventional polar code encoding and decoding. Specifically, it achieves an improvement of 0.94 dB under weak turbulence conditions, a 1.44 dB improvement under medium turbulence conditions, and a 1.25 dB improvement under strong turbulence conditions. Furthermore, at a code rate of $R_2 = 0.5$, comparing the A-BIPCM scheme with conventional polar coding and decoding, significant improvements are observed in various turbulence conditions. Specifically, in the case of weak turbulence, an improvement of 0.96 dB is observed. In the case of medium turbulence, a notable improvement of 1.66 dB is observed. Furthermore, in the case of strong turbulence, a substantial improvement of 1.35 dB is observed. When the decoding width of the A-

BIPCM scheme is increased from 1 to 4 and 8, additional improvements are observed. In weak turbulence, there is a 1.63 dB improvement and a 2.18 dB improvement. In medium turbulence, improvements of 2.37 dB and 3.22 dB are achieved, and in strong turbulence, improvements of 1.913 dB and 2.75 dB are achieved. The simulation results demonstrate that as the turbulence intensity increases, there is a decrease in bit error performance, reflecting the adverse impact of atmospheric turbulence on channel quality. Additionally, under the same turbulent conditions, as the SNR increases, the BER performance improves.

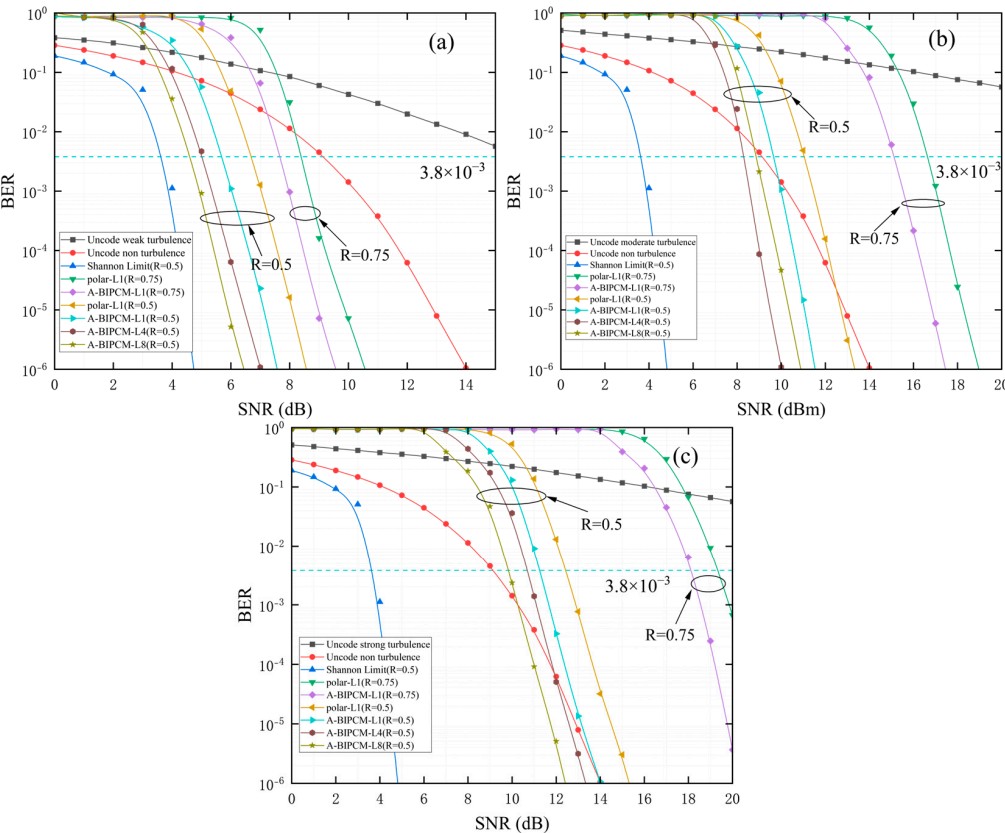

**Figure 2.** FSOC system BER performance curve under different turbulence conditions: (**a**) BER performance curve under weak turbulence conditions, (**b**) BER performance curve under medium turbulence conditions, (**c**) BER performance curve under strong turbulence conditions.

## 4. Experimental Verification Results and Discussion of FSOC System Performance under Weak Turbulence Conditions

To assess the actual performance of the A-BIPCM scheme in combating atmospheric turbulence, this work conducted an experimental verification study on the performance of the FSOC system based on the A-BIPCM scheme. The simulation channel had dimensions of 2 m × 1 m × 0.35 m and was designed to simulate a weak turbulence channel. In this experimental setup, the simulation channel utilized the air between hot and cold plates to create an environment that mimics real atmospheric channels. Convection was employed to replicate the turbulent characteristics of the atmosphere. The intensity of the simulated turbulence was adjusted by controlling the temperature difference between the hot and cold plates, which closely resembles the characteristics of actual channels affected by atmospheric turbulence. The atmospheric coherence length $r_0 = A_{turbulence}\Delta T^{C\Delta}$, an important parameter in characterizing the turbulence effects, $A_{turbulence}$ was measured and determined based on the fitted curve. The linear coefficient is the exponential coefficient of the temperature $\Delta T$ difference between the heating plate and the cooling plate. However, due to limitations in generating turbulence conditions in our laboratory, we can only simulate weak turbulence conditions using the atmospheric turbulence simulation channel.

Figure 3 presents a schematic block diagram of the experimental system used to verify the performance of free-space wireless laser communication employing IM/DD. At the transmitting end of the system, a C-band continuous wave laser with a wavelength of 1550.116 nm and an output power of 10 dBm is utilized. Following polarization controller (PC) adjustment, the output power registers at 7.38 dBm. The polar code encoding modulation signal is generated offline using MATLAB and then loaded into an arbitrary waveform generator (AWG). The AWG has a 3 dB analog bandwidth of 25 GHz and a sampling rate of 32 GSa/s. It is responsible for generating the transmitted RF signal and performs 8 times interpolation, achieving an oversampling rate of 4 GS/s. In the experiment, a microwave amplifier (MA) is employed to amplify the amplitude of the output radio frequency signal from 237 mV to 4.5 V, which is necessary to drive a 10 GHz Mach-Zehnder modulator (MZM). The bias voltage of the MZM is set to 4 V. The optical signal output from the PC is then input into the MZM, and the amplified signal source from the microwave amplifier is added to it. Subsequently, the modulation output of the MZM is directed to the fiber collimator (FC). At the receiving end of the system, the FC is used to couple the received beam into an optical fiber. To control the optical power, a variable optical attenuator (VOA) is employed. The VOA enables real-time adjustment of the optical power of the received signal through attenuation. An avalanche photodiode detector (APD) is utilized for receiving the optical signals. The power input voltage of the detector is set to 12 V, and it has a saturated optical power of $-16$ dBm and the lowest detectable optical power of $-27$ dBm. For signal capture and analysis, a real-time digital storage oscilloscope (DSO) with a sampling rate of 20 GSa/s and a bandwidth of 4 GHz is utilized. The DSO captures the signal during the offline process, allowing for subsequent signal recovery and analysis. Offline digital signal processing (DSP) is performed using MATLAB to recover the transmitted signal. During the offline processing, the start and end positions of the transmission sequence are determined based on the flag bits added to the transmission sequence. The signal is then sampled, and appropriate thresholds are set to identify continuous sampling points as high and low levels of the binary sequence. The binary sequence is deinterleaved and sent to the decoder for decoding operations. The original bit sequence is restored using the Successive Cancellation List (SCL) decoding method combined with CRC. Figure 4 illustrates the experimental system diagram.

Figure 5 depicts the random fluctuation of the time-domain optical signal data over a certain period. In Figure 6, the three conditions represented are as follows: (a) Under certain conditions, where the temperature of the upper and lower plates of the atmospheric turbulence simulation channel is 80 °C and the atmospheric coherence length is $r_0 = 1.839$ at a specific value. In this case, the optical signal experiences minimal jitter in the channel, and the energy of the signal is relatively concentrated and stable. (b) Under different conditions, with the temperature of the upper and lower plates set to 120 °C and the corresponding atmospheric coherence length $r_0 = 1.384$, a slightly stronger fluctuation in the time-domain waveform of the optical signal and a slightly wider dispersion of its energy is indicated. (c) Under further different conditions, where the temperature of the upper and lower plates is set to 160 °C and the corresponding atmospheric coherence length is $r_0 = 0.955$ considered, a greater fluctuation in the time-domain waveform of the optical signal in the weak turbulence channel and a wider dispersion of its energy is indicated. The optical output power jitter curve is shown in Figure 5.

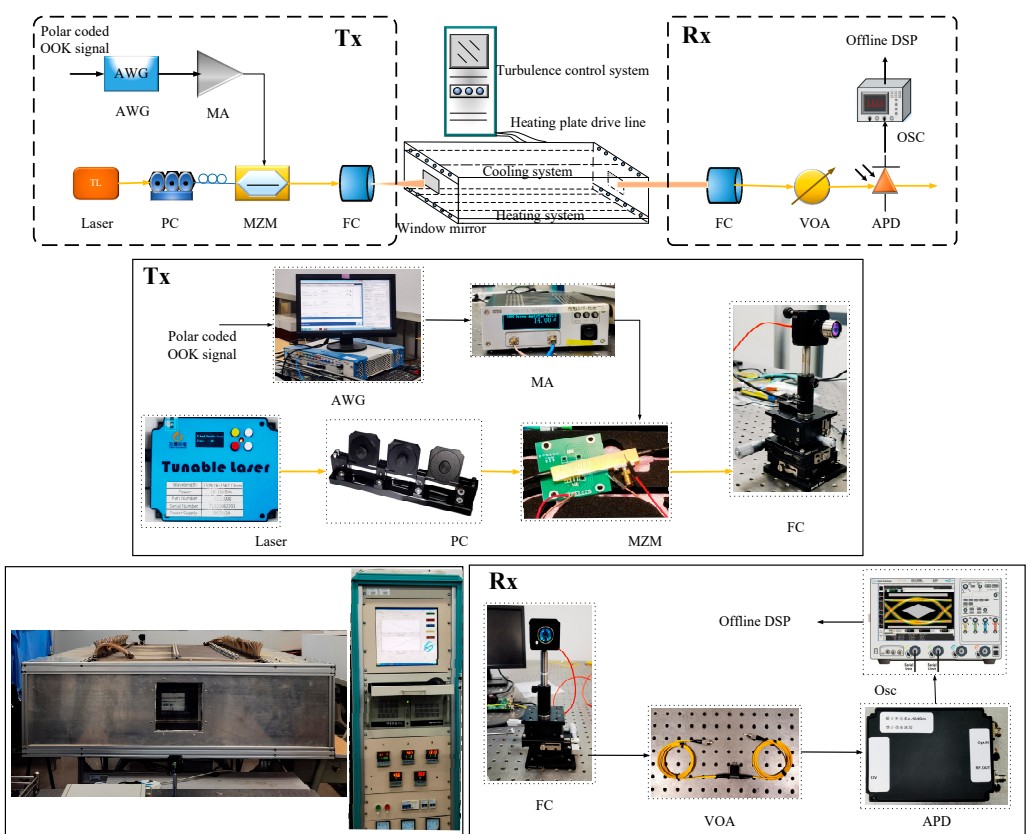

**Figure 3.** Principle block diagram of free space wireless laser communication performance verification experimental system based on IM/DD.

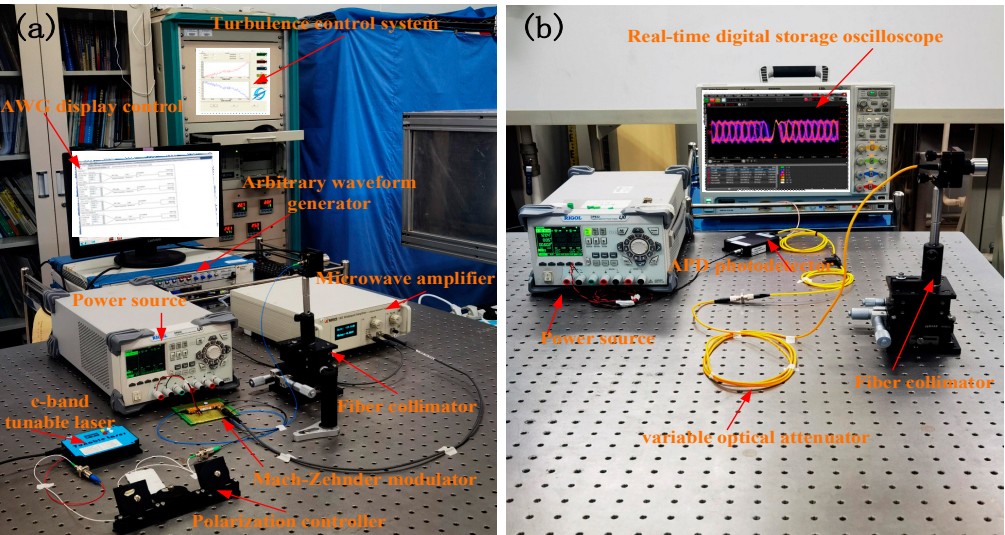

**Figure 4.** Schematic diagram of the transmitter and receiver of the experimental system using adaptive bit-interleaving coding and modulation scheme: (**a**) schematic diagram of the transmitter, (**b**) schematic diagram of the receiver.

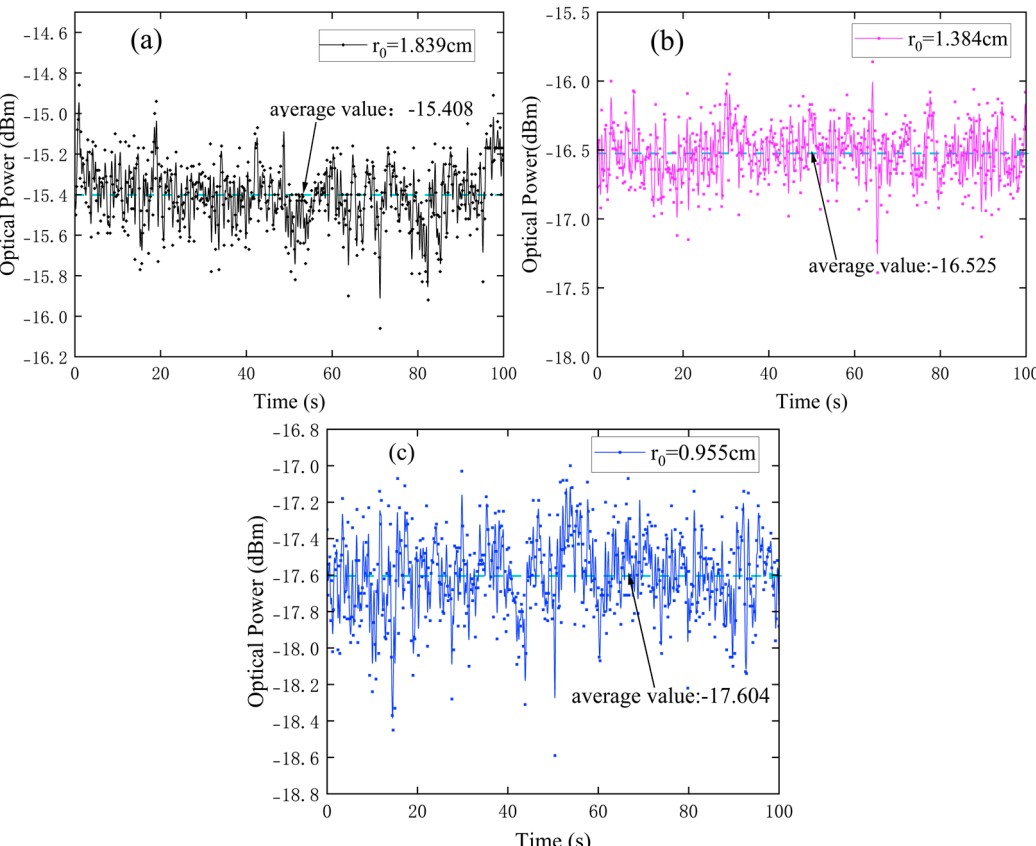

**Figure 5.** Optical output power jitter curve: (**a**) atmospheric coherence length $r_0$ = 1.839, (**b**) atmospheric coherence length $r_0$ = 1.384, (**c**) atmospheric coherence length $r_0$ = 0.955.

Figure 6 presents a statistical analysis of the BER under weak turbulence conditions. The comparison chart depicts the BER performance between the conventional polar code encoding and decoding scheme (represented by the dashed line) and the proposed A-BIPCM scheme (represented by the solid line). In Figure 6a, when the atmospheric coherence length $r_0$ = 0.955, T = 160, the optical power of the detector is $-16$ dBm. The traditional polar code encoding and decoding performance at BER = $9.13 \times 10^{-4}$, whereas the A-BIPCM scheme exhibits a significantly higher performance at BER = $7.02 \times 10^{-5}$. The BER has been improved by representing an order of magnitude. Furthermore, considering the desired BER = $3.8 \times 10^{-3}$ of the system, the conventional polar coding and decoding scheme requires a received optical power greater than $-16.98$ dBm, and the PD-BIPCM scheme requires a received optical power greater than $-17.71$ dBm. Thus, the A-BIPCM scheme achieves an improvement of approximately 0.73 dBm in terms of the required received optical power. In Figure 6b, when the atmospheric coherence length $r_0$ = 1.384, T = 120, the optical power of the detector is $-16$ dBm. The traditional polar code encoding and decoding performance at BER = $6.36 \times 10^{-4}$, whereas the A-BIPCM scheme exhibits a significantly higher performance at BER = $6.82 \times 10^{-5}$. The BER has been improved by representing an order of magnitude. Furthermore, considering the desired BER = $3.8 \times 10^{-3}$ of the system, the conventional polar coding and decoding scheme requires a received optical power greater than $-17.03$ dBm, and the PD-BIPCM scheme requires a received optical power greater than $-18.08$ dBm. Thus, the A-BIPCM scheme achieves an improvement of approximately 1.05 dBm in terms of the required received optical power.

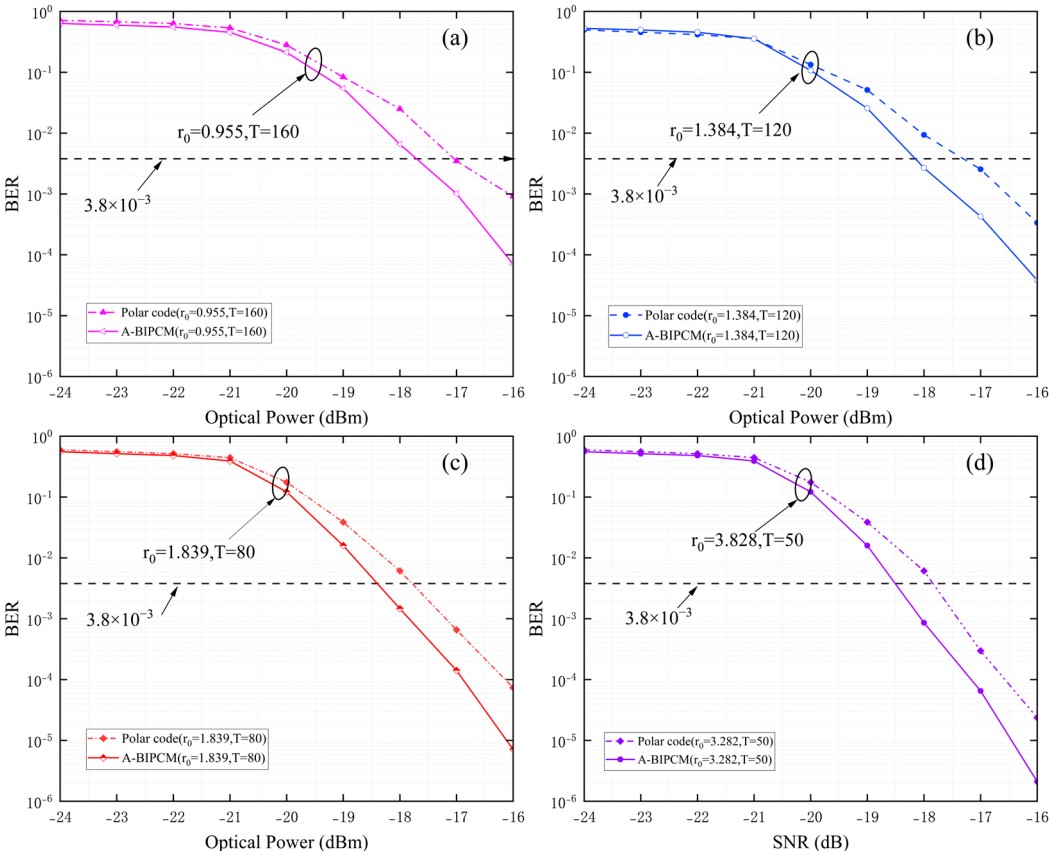

**Figure 6.** BER performance curve of A-BIPCM scheme and traditional polar code encoding and decoding: (**a**) comparison of the BER performance curves of atmospheric coherence length $r_0 = 0.955$, (**b**) comparison of the BER performance curves of atmospheric coherence length $r_0 = 1.384$, (**c**) comparison of the BER performance curves of atmospheric coherence length $r_0 = 1.839$, (**d**) comparison of the BER performance curves of atmospheric coherence length $r_0 = 3.282$.

In Figure 6c, when the atmospheric coherence length $r_0 = 1.839$, T = 80, the optical power of the detector is $-16$ dBm. The traditional polar code encoding and decoding performance at BER = $7.13 \times 10^{-6}$, whereas the A-BIPCM scheme exhibits a significantly higher performance at BER = $7.36 \times 10^{-5}$. The BER has been improved by representing an order of magnitude. Furthermore, considering the desired BER = $3.8 \times 10^{-3}$ of the system, the conventional polar coding and decoding scheme requires a received optical power greater than $-17.69$ dBm, and the PD-BIPCM scheme requires a received optical power greater than $-18.08$ dBm. Thus, the A-BIPCM scheme achieves an improvement of approximately 0.73 dBm in terms of the required received optical power. In Figure 6d, when the atmospheric coherence length $r_0 = 3.282$, T = 50, the optical power of the detector is $-16$ dBm. The traditional polar code encoding and decoding performance at BER = $2.36 \times 10^{-5}$, whereas the A-BIPCM scheme exhibits a significantly higher performance at BER = $2.11 \times 10^{-6}$. The BER has been improved by representing an order of magnitude. Furthermore, considering the desired BER = $3.8 \times 10^{-3}$ of the system, the conventional polar coding and decoding scheme requires a received optical power greater than $-18.42$ dBm, and the PD-BIPCM scheme requires a received optical power greater than $-17.69$ dBm. Thus, the A-BIPCM scheme achieves an improvement of approximately 0.71 dBm in terms of the required received optical power. Under weak turbulence conditions, the A-BIPCM scheme can provide better BER performance compared with traditional polar code encoding and decoding. In addition, as the received optical power increases, the BER shows a downward trend and increases by one order of magnitude. This shows that atmospheric turbulence has a significant impact on channel quality, and increasing

transmit power can improve the information transmission quality of atmospheric optical channels.

## 5. Conclusions

This work employs polar spectrum and average PEP as analytical tools to investigate the upper bound of the error probability, enabling the proposition of an adaptive bit-interleaved polar coding modulation (A-BIPCM) approach. The theoretical model demonstrates that, within the simulated atmospheric channel established using the Monte Carlo model, this coding method effectively mitigates the impact of attenuated atmospheric conditions, as evidenced by its superior performance in terms of BER. Furthermore, an experimental Free Space Optical Communication (FSOC) setup incorporating a controllable atmospheric turbulence channel was implemented. When T = 160, when the optical power of the detector is $-16$ dBm, BER = $9.8 \times 13^{-4}$, traditional polar code encoding and decoding, A-BIPCM scheme BER = $7.02 \times 10^{-5}$. A comparison between the A-BIPCM method and the polar code encoding and decoding scheme without adaptive bit-interleaving revealed a one-order-of-magnitude improvement in BER performance for the former, consistent with the simulation results. This substantiates that the proposed A-BIPCM method possesses enhanced error correction capabilities, lower computational complexity, and effectively addresses the issue of a persistent long string of bit errors caused by turbulence.

**Author Contributions:** Conceptualization, Q.J. and Z.L. (Zhi Liu); methodology, Q.J.; software, Q.J.; validation, H.Y. and Q.J.; formal analysis, Q.J.; investigation, Z.L. (Zhonglin Luo), C.C., H.L. and G.J.; resources, X.Z.; data curation, Q.J.; writing—original draft preparation, Q.J.; writing—review and editing, Z.L. (Zhi Liu); visualization, S.L.; supervision, Z.L. (Zhi Liu) and P.L.; project administration, Z.L. (Zhi Liu); funding acquisition, Z.L. (Zhi Liu). All authors have read and agreed to the published version of the manuscript.

**Funding:** National Natural Science Foundation of China Ye Qisun Science Foundation, grant number U2141231. China Ye Qisun Science Foundation totals 2.6 million.

**Institutional Review Board Statement:** Not applicable.

**Informed Consent Statement:** Not applicable.

**Data Availability Statement:** Data are contained within the article.

**Conflicts of Interest:** The authors declare no conflicts of interest.

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
