# Peer review of "Performance of Adaptive Bit-Interleaved Polar Coded Modulation in FSOC System"

_photonics, doi:10.3390/photonics11010034_

Round 1

Reviewer 1 Report

Comments and Suggestions for Authors

Please take note of the following revision suggestions:

1. In the Introduction, it is recommended to include specific performance metrics when describing the results.

2. It is advised to streamline the theoretical introduction in the second section to avoid the third section's simulation results appearing insufficient.

3. Ensure that the use of formulas aligns with the experimental results and remains consistent with the simulation results.

4. Consider modifying the way of marking legends in Figure 2.

5. It is suggested to supplement the Conclusions section with quantitative results.

Comments on the Quality of English Language

NA

Author Response

Thank you very much for your review comments and valuable suggestions on our paper. We greatly appreciate your careful review and guidance. Based on your comments, we have made detailed revisions to the paper and addressed the issues you raised. Here are our responses to your comments:

Please take note of the following revision suggestions:

  1. In the Introduction, it is recommended to include specific performance metrics when describing the results.

Thank you very much for your suggestion, We have added specific performance metrics to the Introduction and ensured that the performance indicator descriptions are more specific and clear.

  1. It is advised to streamline the theoretical introduction in the second section to avoid the third section's simulation results appearing insufficient.

We have simplified the theoretical introduction in Part 2, At the same time, relevant content has been appropriately supplemented. To support the simulation results part of the third part, ensure that the focus of the paper is more prominent.

  1. Ensure that the use of formulas aligns with the experimental results and remains consistent with the simulation results.

We have carefully checked the use of the formula and ensured that it matches the experimental results. In addition, we further verified the consistency between the formula and the simulation results.

  1. Consider modifying the way of marking legends in Figure 2.

We have modified the way legends are labeled in Figure 2 to make them clearer and easier to understand.

  1. It is suggested to supplement the Conclusions section with quantitative results.

We have added more quantitative results to the Conclusions section to strengthen the summary and quantitative evaluation of research results.

Reviewer 2 Report

Comments and Suggestions for Authors

This manuscript proposes an adaptive bit-interleaved polar coding modulation (A-BIPCM) method for moderating BER which caused by atmospheric turbulence disturbance, and it seems interesting. However, there are some issues that need to be addressed:

1. What are the advantages of bit-interleaved polar coding modulation over existing BICM? How is the adaptiveness of the A-BIPCM FSOC system depicted in the system diagram?

2. How is the relationship established between turbulence intensity and the choice of interleaving methods for each channel block? It is recommended that the authors provide a more in-depth analysis and explanation in the paper.

3. It is recommended to enhance the Introduction by providing specific performance indicators for the results to improve accuracy and clarity.

Comments on the Quality of English Language

minor revision

Author Response

Thank you very much for your review comments and valuable suggestions on our paper. We greatly appreciate your careful review and guidance. Based on your comments, we have made detailed revisions to the paper and addressed the issues you raised. Here are our responses to your comments:

This manuscript proposes an adaptive bit-interleaved polar coding modulation (A-BIPCM) method for moderating BER which caused by atmospheric turbulence disturbance, and it seems interesting. However, there are some issues that need to be addressed:

  1. What are the advantages of bit-interleaved polar coding modulation over existing BICM? How is the adaptiveness of the A-BIPCM FSOC system depicted in the system diagram?

Under different turbulent conditions, the A-BIPCM method uses minimum logarithmic upper limit weight (MLUW) for adaptive bit interleaving. Compared with polar code encoding and decoding, it achieves significant performance improvement and can effectively reduce long-chip error interference. Improve the reliability and bit error rate performance of the system, which is further explained in Section 2.2. In order to clearly describe the adaptability of the A-BIPCM FSOC system in the system diagram, a schematic sequence diagram of the interleaved part is supplemented in Figure 1.

  1. How is the relationship established between turbulence intensity and the choice of interleaving methods for each channel block? It is recommended that the authors provide a more in-depth analysis and explanation in the paper.

We have conducted a more in-depth analysis and explanation of the relationship between turbulence intensity and selection, and provided relevant detailed instructions in the second part of the paper to ensure that readers have a clearer understanding of the establishment of the staggered method.

  1. It is recommended to enhance the Introduction by providing specific performance indicators for the results to improve accuracy and clarity.

We have strengthened the paper's presentation by providing concrete performance and provided indicators of the accuracy and clarity of the results, To ensure readers have a more comprehensive understanding of the research results.

Reviewer 3 Report

Comments and Suggestions for Authors

Review of manuscript Photonics-2761862

Performance of adaptive bit interleaved polar coded modulation in FSOC system

By Qingfang Jiang, Zhi Liu, Haifeng Yao, Xin Zhng, Shutong Liu, Chenming Cao, Gang Jing, Hao Li, Peng Lin

The authors investigated theoretically and experimentally a novel adaptive bit-interleaved polar coded modulation (A-BIPCM) method in a free space optical communication (FSOC) system. This method permits to combat the fading effects and long string of bit errors interference caused by the atmospheric turbulence. The results obtained demonstrate a substantial improvement of the FSOC system performance as compared to conventional polar code encoding and decoding. The proposed manuscript is interesting and well organized. It contains novel results. The experimental implementation of the new method is especially important. The paper can be interesting for researchers and engineers occupied in the FSOC systems.

 However, the proposed manuscript cannot be accepted for publication in the present form. The following minor revisions are necessary.

1.      Page 3, equation (1). The coefficient  is not defined.

2.      Page 3, equation (2). The parameters  are not defined. It is unclear whether the known expression is used by the authors, or they derived this expression. In the first case the corresponding reference is necessary. Otherwise, the authors must briefly explain the derivation process.

3.      Page 4, equations (5) and (6). The references for the sources of these expressions are absent. The is not defined. Is it the polar code mentioned in line 158?

4.      Figures 3 and 5 are too small. The details and captions inside the Figures are hardly visible.

5.      There are some misprints in the text. See, for instance, page 1, Abstract, lines 16-17. “In this paper, Adaptive bit-interleaved polar coded modulation (A-BIPCM) method based on minimum logarithmic upperbound weight (MLUW) are introduced…”. Page 1, Introduction, lines 34-35. “Free-space optical communications (FSOC), also known as optical wireless communication, offers advantages…”. Page 2, lines 46-48.  “ To further enhance the system’s performance , bit-interleaved coded modulation (BICM) is introduced as a communication scheme in FSOC, Initially proposed by Zehavi,…”. Page 7, line 245. “Specifically, Under weak turbulence conditions…”.

Comments on the Quality of English Language

Moderate editing of English language is necessary. Some misprints are mentioned in the Review. 

Author Response

Thank you very much for your review comments and valuable suggestions on our paper. We greatly appreciate your careful review and guidance. Based on your comments, we have made detailed revisions to the paper and addressed the issues you raised. Here are our responses to your comments:

The authors investigated theoretically and experimentally a novel adaptive bit-interleaved polar coded modulation (A-BIPCM) method in a free space optical communication (FSOC) system. This method permits to combat the fading effects and long string of bit errors interference caused by the atmospheric turbulence. The results obtained demonstrate a substantial improvement of the FSOC system performance as compared to conventional polar code encoding and decoding. The proposed manuscript is interesting and well organized. It contains novel results. The experimental implementation of the new method is especially important. The paper can be interesting for researchers and engineers occupied in the FSOC systems. However, the proposed manuscript cannot be accepted for publication in the present form. The following minor revisions are necessary.

  1. Page 3, equation (1). The coefficient is not defined.

We have given an explanation of the relevant parameters in equation (1) on page 3, and have made corresponding corrections in the paper.

  1. Page 3, equation (2). The parameters ,are not defined. It is unclear whether the known expression is used by the authors, or they derived this expression. In the first case the corresponding reference is necessary. Otherwise, the authors must briefly explain the derivation process.

The relevant parameters of equation (2) on page 3 have been defined, Equation (2) uses known expressions and has been supplemented in the article.

  1. Page 4, equations (5) and (6). The references for the sources of these expressions are absent. The is not defined. Is it the polar code mentioned in line 158?

On page 4, the source references for the expressions in equations (5) and (6) have been cited, and the parameters of the expressions have been defined to eliminate ambiguity. These expressions are related to the polar code mentioned on line 158.

  1. Figures 3 and 5 are too small. The details and captions inside the Figures are hardly visible.

Figures 3 and 5 have been modified for this article to ensure that their size and detail are clearly visible, and the titles have been adjusted accordingly.

  1. There are some misprints in the text. See, for instance, page 1, Abstract, lines 16-17. “In this paper, Adaptive bit-interleaved polar coded modulation (A-BIPCM) method based on minimum logarithmic upperbound weight (MLUW) are introduced…”. Page 1, Introduction, lines 34-35. “Free-space optical communications (FSOC), also known as optical wireless communication, offers advantages…”. Page 2, lines 46-48. “ To further enhance the system’s performance , bit-interleaved coded modulation (BICM) is introduced as a communication scheme in FSOC, Initially proposed by Zehavi,…”. Page 7, line 245. “Specifically, Under weak turbulence conditions…”.

We have corrected misprints in the text, including lines 16-17 of the abstract on page 1, lines 34-35 of the introduction on page 1, lines 46-48 of page 2, and line 245 of page 7 . These corrections will ensure accuracy and clarity in the text.
